# Evidence of a cascading positive tipping point towards electric vehicles

Jean-François Mercure [1,2,3,5] ✉, Aileen Lam [2,4,5], Joshua E. Buxton [3], Chris A. Boulton [3], Amir Akther[3] & Timothy M. Lenton [3]

Electric vehicles have recently seen rapid innovation, decline in cost and a rise in popularity. Past a tipping point where uptake becomes self-propelling, electric vehicles could irreversibly replace internal combustion engine vehicles, as industry discontinues conventional production chains. Here we provide evidence that this tipping point has occurred or lies within the next few years in lead markets of the European Union and China, and potentially the United States, which could spill out into peripheral vehicle markets across the rest of the world. The historical evidence shows a sudden decline in conventional vehicle sales starting around 2019 concurrent to a rapid rise in sales of electric vehicles. Critically, we observe a loss of resilience of the incumbent technology consistent with the approach to a tipping point. We use simulations of technology evolution to identify timescales for cost-parity and policy frameworks that could accelerate the transition to largely eliminate combustion vehicles before 2050.

Road transport emissions are rising rapidly, due to growing fleets worldwide, despite fuel efficiencies increasing in most vehicle markets[1]. Private passenger road transport generates 45% of transport emissions and 30% of its growth, the transport total accounting for 22% of global $CO_2$ combustion emissions[2], something that requires urgent attention from policy-makers. Since the late 2000s, electric vehicles (EVs) have been seen as a key solution for decarbonising private passenger road transport[3]. Plug-in hybrid electric vehicles (PHEVs) overcome issues of range anxiety perceived with EVs[4,5], but they are now to be phased out along with internal combustion engine vehicles (ICEVs) in the European Union (EU) and the UK by 2035, whereas the ranges of EVs are improving as a result of battery innovation. EVs have seen their costs decline drastically following large scale investment, anticipating or benefiting from favourable policy environments, particularly in the world's leading car markets of Europe, the United States and China[6–18]. Hence, it is important and timely to examine whether the observed diffusion of EVs could become self-propelling and tip towards a state of dominance.

In complex systems, a 'tipping point' occurs when a small change triggers a large, non-linear change of state of a system, supported by self-propelling (strong, positive) feedback[19–21]. Technologies often see tipping points in their diffusion processes where there are increasing returns, as costs decline due to increasing investment anticipating rising sales, and sales increase due to declining costs and improving consumer accessibility[20,22–24]. This has occurred before in the early 1900s with the onset of oil-based mobility[25–27]. It could conceivably occur with EVs once ownership costs and the diversity of models available on markets become comparable with incumbent alternatives, and various user adjustment hurdles are overcome such as range anxiety[28–30].

Technological change occurs temporally but also spatially, from a core location where innovation is done, towards a rim and a periphery, where later adopters benefit from the experimentation done earlier within the core[31–35]. We thus expect that reaching a critical mass and regional tipping points in the leading core markets (henceforth 'lead markets'[36]), including Europe, China, and the US, could trigger the spread of a global EV tipping point.

Statistical indicators of resilience loss of an incumbent state prior to a tipping point are detectable in climatic and ecological systems[37]. The same methods may be applicable to rapid technological

[1]The University of Exeter Business School, Exeter, UK. [2]The World Bank, Washington, DC, USA. [3]Global Systems Institute, University of Exeter, Exeter, UK. [4]Department of Economics, Faculty of Social Sciences, University of Macao, Macau, China. [5]These authors contributed equally: Jean-François Mercure, Aileen Lam. ✉e-mail: j.mercure@exeter.ac.uk

transitions, where we term them 'early opportunity signals'. These may show elements of the economy which are losing resilience and thus approaching a tipping point to transition to a new state, in the present case, one dominated by EV ownership. The rationale is that a generic dynamical systems behavior of weakening of overall damping negative feedback (leading eigenvalue increasing towards zero, termed 'critical slowing down'[37]) occurs before a tipping point. These methods have been used to show for instance, that a tipping point in the public's view of EVs may be approaching in the UK[38].

Earlier work studied the diffusion of EVs using a highly empirical conditional technological forecasting model, called 'Future Technology Transformations' applied to transport (FTT:Transport), to simulate the future evolution of those markets under different policy contexts. FTT simulates the commonly observed coupled self-reinforcing S-shaped diffusion profile and the experience curve, both calibrated to data (See Methods and Supplementary Notes 1–2 for model definitions)[39–42]. It can be used to identify, within some uncertainty, when price and cost parity with ICEVs may be achieved globally, which is a necessary (although not sufficient) condition for a tipping point (Supplementary Note 3).

In this work, we show empirically that a tipping point towards EVs is occurring in two of three leading markets, and could spill out into a global tipping point. We explore the evolution and diversity of both EV and ICEV car markets and their evolution in recent years, including their diversity and cost trajectories, in the three lead markets (Europe, the US, China) and one peripheral market (India). We show empirically that the ICEV system in Europe and China is losing resilience, and a self-propelling transition to EVs is approaching. These trends, however, are not sufficiently rapid to achieve stated net-zero emissions targets. Hence, we use FTT:Transport to confirm these findings, and to identify policy mixes that can achieve climate targets. Lastly, we determine the pace at which EV policy support in lead markets could produce cost parity in peripheral markets, allowing the tipping point to spill out worldwide. Coordinated EV policy action in lead markets could therefore be instrumental to trigger a global electric mobility tipping point, even in developing nations in which policy space may be limited. The contribution of this paper is fourfold: (1) it assembles a comprehensive dataset 2016-2023 that tracks the evolution of leading vehicle markets to identify the tipping point; (2) it applies the early warning tipping point signals method to vehicle data to confirm the existence of the tipping point; (3) it calibrates FTT:Transport to this data timeseries to make predictions of cost parity and policy effectiveness; (4) it establishes empirical relationships between EV costs, prices and deployment.

## Results
### Context and theory
Two main empirically observed mechanisms drive technological diffusion. The first is the diffusion mechanism, first observed with agricultural innovations, in which early adopters create a demonstration effect that facilitates later adoptions[22,32,43,44]. This produces an S-shaped diffusion profile over time (Fig. 1a). The second is the experience curve, in which production costs of technologies decline with increasing experience (Supplementary Figs. 1–3)[23,45]. This produces a power law cost function of cumulative production. As is typically observed during technology transitions, diffusion processes cause cost declines, and cost declines accelerate diffusion[39,46–48]. The resulting non-linear dynamical system generates potential branching points in which temporary early-stage technology pushes can cause technological avalanches, or tipping points, if initial high-cost hurdles are overcome and performance overtakes competitors. This appears to have occurred with solar energy[47,49]. Cost reductions become limited, however, as it takes an exponentially rising innovative effort to maintain a constant rate of improvement.

When such non-linear dynamical systems approach a tipping point, fluctuations in state variables typically increase in magnitude and duration, due to the phenomenon of 'critical slowing down'[37]. In essence, as overall damping (negative) feedbacks get weaker and amplifying (positive) feedbacks get stronger, small perturbations generate larger responses that take longer to recover from. This can be visualised as a shallowing and broadening of the potential well that describes the initial 'incumbent' state of the system, allowing a growth in the magnitude of fluctuations and a slowing of their recovery, which diverges at the transition point (Fig. 1b). Characterising fluctuations in timeseries can thus be used to search for early signals before a transition occurs, as observed in some dynamical properties of the Earth system[37, 50].

In a technological transition, fluctuations can be expected to be observed over the technological composition as increasing numbers of agents (consumers, investors) begin to switch their preferences influenced by their peers, such as observed with sales and other metrics of consumer attention[20,38]. This is consistent with the qualitative literature in which regime weakening leads to market instability and rapid and sudden transitions to new socio-technical regimes[27].

The dynamics of technological evolution can also be simulated into the future, conditional on cost reductions continuing to be a function of cumulative investment, industry growth remaining unhindered, and other contextual elements remaining the same (e.g. public policy, interest rates, demand growth, social order, availability of inputs to production). We do this with the FTT model (see Supplementary Note 4), in which we use observed diffusion trajectories for 32 countries and rates of cost reductions due to fleet expansions in five types of power trains and 3 engine/motor classes each, for four key regions, to parameterise a coupled dynamical diffusion system that we simulate until 2050. The methods section describes the steps we use to identify a tipping point.

### Evidence of a tipping point from historical data
Figure 2 shows the ongoing exponential growth of the global EV and PHEV fleet with a doubling time of ~1.5 years, concurrent with rapid declines in battery prices (see Methods for data sources)[6–9,11–16,51,52]. The cost of batteries, which is the single most expensive component of making EVs, has gone down by over 85% since 2010, as a result of cumulative process innovation. This has been driven by rising investments, notably by Tesla in the US, followed by diverse companies in China and most European manufacturers. EV cost reductions occur not only within the realm of batteries but also extend to other components and advancements in EV production platforms. We observe that the majority of EV sales occur in three lead markets where most of the innovation also occurs, China, the US and Europe. We observe exponentially rising EV sales in at least 32 countries worldwide (Supplementary Fig. 5), and doubling times for the EU, China, and US of 1.3, 1.0, and 1.7 years.

Figure 3 shows the rapidly evolving distribution of sales of 2452 tracked models in our study between 2016 and 2023, in the lead markets plus a key peripheral market, India. We observe both an exponential rise in the variety, sales, and availability of EVs and PHEVs, against a relatively sudden decline in sales and variety of ICEVs. The decline appears influenced by the COVID-19 pandemic, but sales of ICEVs did not recover with the rest of these economies after the start of 2021, while the sales of EVs and PHEVs do not appear affected by the pandemic. Subsequently, there has been a partial recovery of ICEV sales in some markets (but not Europe).

The distribution of sales also changed shape over time for EVs, where variety has grown in Europe (including individual EU countries), the US, and China, at a rate of more than 30% annually, while variety for ICEVs peaked and declined everywhere (Supplementary Fig. 6). With many if not most automakers having made announcements to develop and produce EVs on a larger scale in the coming years[6], the number of

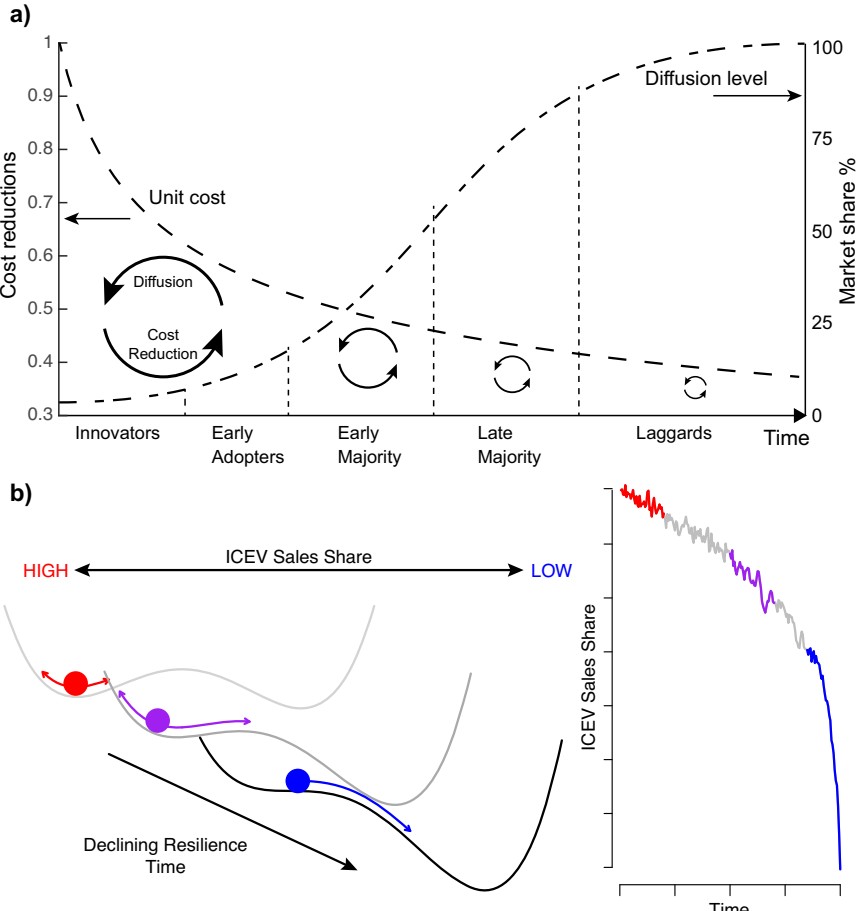

**Fig. 1 | Diagram illustrating the self-reinforcing S-curve and technological tipping points. a** Visual representation of the diffusion curve interacting with the learning curve at different stages of adoption. **b** A visual representation of the system approaching a tipping point, moving from one state (high ICEV sales share) to another (low ICEV sales share). The ball-in-the-well representations (left) show the system losing resilience over time, from red, to purple, to blue, eventually tipping. The time series (right) shows the system approaching and just passing the tipping point. ICEV stands for Internal Combustion Engine Vehicle.

models can be expected to continue to increase rapidly in the lead markets. Rising EV variety against declining ICEV variety testifies to a transition towards EVs occurring within vehicle manufacturing, as it facilitates the diffusion of EVs and curtails the dominance of ICEVs across market segments (Supplementary Note 5).

Figure 4 shows evidence of loss of resilience in the measured fluctuations over time in sales of ICEVs in Europe and China (see Methods). Prior to the abrupt decline in ICEV market shares in both markets at the start of 2020, there was a visible marked increase in the magnitude and persistence of fluctuations and a corresponding increase in lag-1 autocorrelation (AR(1)) and variance of the ICEV market share. This suggests a loss of resilience of sales of the incumbent technology (ICEVs), consistent with heading towards a potential tipping point in the market. These early opportunity signals are observed in a summation of the largest European country markets (Germany, the UK, and France) and China, with AR(1) and variance showing strongly increasing trends (high Mann-Kendall Tau values). The US market shows no consistent signal of loss of resilience prior to 2020, consistent with the lack of an abrupt decline in ICEV market share thereafter.

**Modelling future vehicle markets under policy inputs**

Given these empirical indicators that major markets are heading towards a tipping point to EVs, we now turn to our empirically-grounded, process-based simulations of the relevant dynamical social-technical transition system using the FTT:Transport model (see

Methods and Supplementary Note 2). We first focus on projecting the current technological trajectory of vehicle systems, in order to estimate the timescales involved in the overall transition, and monitor as proxies for tipping points the moments at which Total Cost of Ownership (TCO) parity and purchase price parity occurs. We also monitor the diffusion levels and test for positive feedbacks. We do this for the entire world, modelling diffusion in 32 individual countries, and broader regions covering the rest of the world.

Our simulations provide projections, with uncertainty ranges, for EV prices and TCO, along with their ICEV counterparts, for three market segments (Supplementary Fig. 7). Uncertainty ranges encompass plausible values for experience curve rates, discount rates, and critical mineral prices used for manufacturing batteries. For consumer discount rates of 20% or less and average intensity of use, TCO parity has already been achieved in the lead markets for most EV types. According to our simulations, TCO parity can also be expected to happen soon in the rest of the world, with many developing countries expected to break cost parity between 2025 and 2030, contingent on trade policy remaining the same for EVs as for ICEVs (Supplementary Note 6).

On the basis that EVs generally cost more upfront but less over the lifecycle than ICEVs, price parity can be expected to occur later than TCO parity. In our simulations, in the absence of further changes of policy, EVs in Europe and China achieve price parity with ICEV between 2025 and 2028. This will be followed by the US, Canada, and South Korea between 2026 and 2030. The rest of the world follows as the

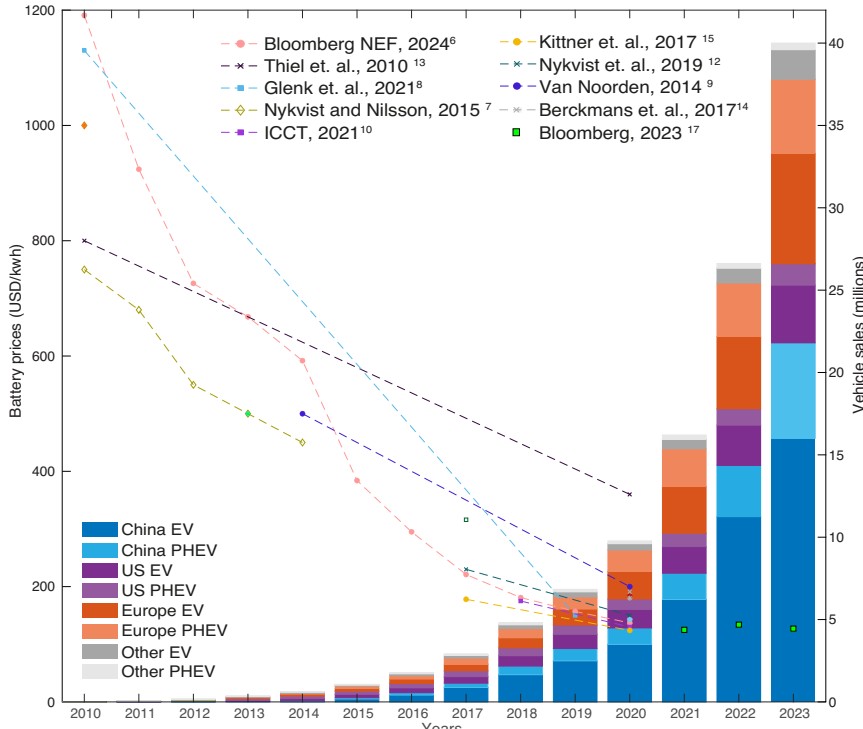

**Fig. 2 | Diffusion history for electric vehicles.** Rapid rise of EVs and PHEVs in the major car markets and worldwide (right axis and bars) and concurrent rapid battery cost reductions (left axis and lines). EV stands for Electric Vehicle, while PHEV stands for Plug-in Hybrid Electric Vehicle.

price falls and more models become available. EVs in all other global regions break initial price parity with ICEVs between 2030 and 2035. These FTT:Transport results are consistent with our purely empirical results that Europe and China are approaching a tipping point to EVs ahead of the US and the rest of the world. Fig. 5a shows that the existence of a tipping point is further confirmed by simulations in which under current policies (i.e., no additional 'forcing' of the system), EVs come to dominate car fleets by 2050 in Europe and China, but not in the US nor India.

Figure 5 also explores how public policy can accelerate the EV transition and bring about a tipping point in markets where it may not otherwise occur. Figure 5a displays the baseline evolution of the technology composition of fleets in the three lead markets plus India, showing the current transition momentum resulting from existing and past policy and the cumulative effect of EV innovation (Supplementary Fig. 8 shows this for 30 countries). The shares of EVs grow in each market, making costs decline further, but the current diffusion pace is too slow to meet climate targets. Despite the existence of a tipping point in the EU and China, policies bringing it forward and supporting a faster transition are essential to implement in order to achieve climate targets in time. Supplementary Tables 8–19 provide policy details and sensitivity analyses.

The second to fourth columns of Fig. 5b–d show the impacts of sequentially adding fuel taxes and EV subsidies, fuel economy regulations, and EV mandates to the policy mix (See Methods for definitions and Supplementary Note 7 for sensitivity analyses). This combination greatly accelerates the transition and in the model is sufficiently rapid to meet road transport's share of net-zero targets in 2050. On their own, taxes and subsidies are insufficient in the model, since fuel or vehicle taxes and EV subsidies have limited effect in situations where the availability of EVs is itself limited in the early stages of diffusion. We also find[40] that EV mandates on their own have a similarly muted impact in the model, since increasing the availability of EVs does not guarantee sales if they are not financially attractive. Fuel economy regulations help accelerate the turnover of old, inefficient

vehicles and reduce emissions, but on their own have limited potential in the model since they do not directly support EVs. Combining all three instruments has a defining impact on emissions and on the diffusion of EVs in the model, since the introduction of EV mandates, by increasing availability, magnifies the effectiveness of taxes and subsidies at influencing consumer choices of vehicles, while regulations reduce emissions in the interim period. Our projections of cost/price parity (Supplementary Fig. 7) can help determine when EV subsidies can safely be discontinued.

We stress that while uncertainty post-2035 is substantial, once ICEV sales go below 50%, their decline becomes self-propelling[27], and therefore could accelerate faster than our model predicts, due to diseconomies of scale, decline in infrastructure for fueling and maintenance, making them less convenient, and ultimately discontinued production chains, which we do not model explicitly.

### The plausibility of a global EV tipping point and limitations of this study

The tipping point process has begun to spread to peripheral markets, induced by the lead markets, notably in developing markets where policy space and policy-making capacity could be limited. Figure 6 shows how historically EV sales have kicked-off in core lead markets and gradually spilled out to neighbouring markets (see Supplementary Fig. 9 for comparable maps of growth rates). We find that estimated timelines for EV price parity in India, induced by action in the various lead markets, can be brought forward by between 3 and 5 years, considering various uncertainties over experience curve rates and mineral prices (Supplementary Fig. 10). These pieces of evidence together suggest that concerted action by the lead markets can bring forward in time EV price parity in peripheral markets, and a spilling out effect from core to peripheral markets.

Cost parity is, however, not necessarily sufficient, from a technology diffusion perspective, to trigger an EV transition in all peripheral markets and the rest of the world. For instance, the availability of EVs could remain patchy or even absent in many countries. However,

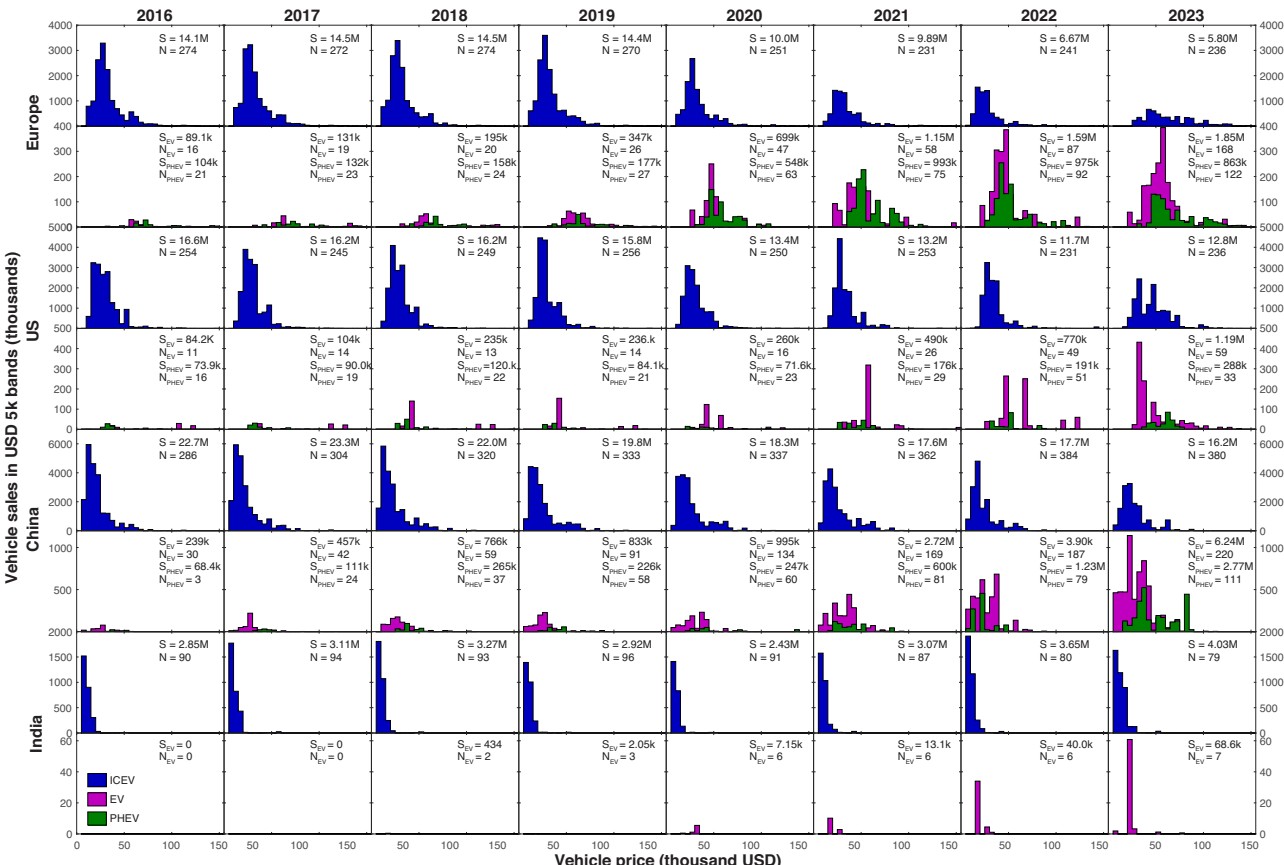

**Fig. 3 | Evolution of vehicles markets and prices.** Price distribution time series for conventional petrol/diesel vehicles, EVs and PHEVs in Europe, China, the US, and India. Panels indicate total vehicle numbers and variety. S, $S_{EV}$ and $S_{PHEV}$ and N, $N_{EV}$ and $N_{PHEV}$ refer to sales (S) and numbers of models available (N) of conventional vehicles, electric (EV) and plug-in hybrid vehicles (PHEV) respectively.

since fleets in some countries are dominated by used cars exported from richer countries, affordable used EVs from the lead markets can eventually become available in the peripheral markets, reinforcing the impact of action in lead markets on peripheral markets. EV mandates and ICEV phase-out regulations in peripheral markets will be important to incentivise foreign vehicle dealers and manufacturers to place EVs there, which, along with price parity, could allow a transition to unfold.

Against all these factors contributing to a continued decline in the cost of EV production, rising scarcity[53] and costs of battery raw materials—especially cobalt, nickel, and lithium—have caused increases and fluctuations in battery prices. For instance, despite the continued decrease in the cost of low-carbon technologies due to their increasing adoption, 2022 marked the first recorded increase in battery prices in 12 years[17], raising concerns of delays in achieving price parity. However, following unprecedented price increase, battery prices have continued to fall in 2023[17]. Our findings indicate that EVs in all segments remain projected to attain price parity between 2025 and 2030 even under high mineral prices scenarios (Methods and Supplementary Note 7 for sensitivity analyses).

Lastly, lack of charging infrastructure can act as a barrier to the diffusion of EVs in the early stages of diffusion, particularly in developing countries. We find evidence that the charging infrastructure has up to now co-evolved with the diffusion of EVs, where the causation goes both ways (Supplementary Figs. 11, 12, see Methods). In countries with large EV shares, the number of charging points per EV has generally started off at between 0.5 and 1.5 in the early stages of diffusion before stabilising towards 0.2 in the later stages. This suggests that

improving publicly funded infrastructure initially accelerates the diffusion of EVs, which itself induces further privately financed infrastructure expansion later.

## Discussion

Our data and analysis provide evidence that several European markets and China have begun tipping towards EV dominance, although this is not convincingly the case for the US. That EV sales are rising exponentially while those of ICEVs are declining indicates that manufacturers worldwide have begun to discontinue many ICEV models while marketing increasing numbers of EV models (Supplementary Note 8). Rising variety in EVs concurrent with declining variety in ICEVs is also a key indicator of tipping behaviour, suggesting a switch in commitment of manufacturers from ICEVs towards EVs.

Furthermore, our analysis identifies a weakening of the ICEV regime through a loss of resilience, as is expected prior to a tipping point. The FTT:Transport model results confirm that the system, exhibiting two alternative stable states, is headed towards a tipping point. Hence the major markets of China and Europe appear, according to our empirics and our model, to be on the self-propelling path to transitioning towards the EV-dominated state. Other major markets, notably the US and India, have the potential for such a tipping point in future, which can be brought forward by coordinated public policies. Ultimately, as EV prices continue to fall and variety continues to increase, there is the potential for the tipping point to EVs to cascade to all car markets worldwide. Scale expansion in emerging markets drives costs down even further back in the lead markets. This mechanism can become critical to achieving a zero-carbon mobility

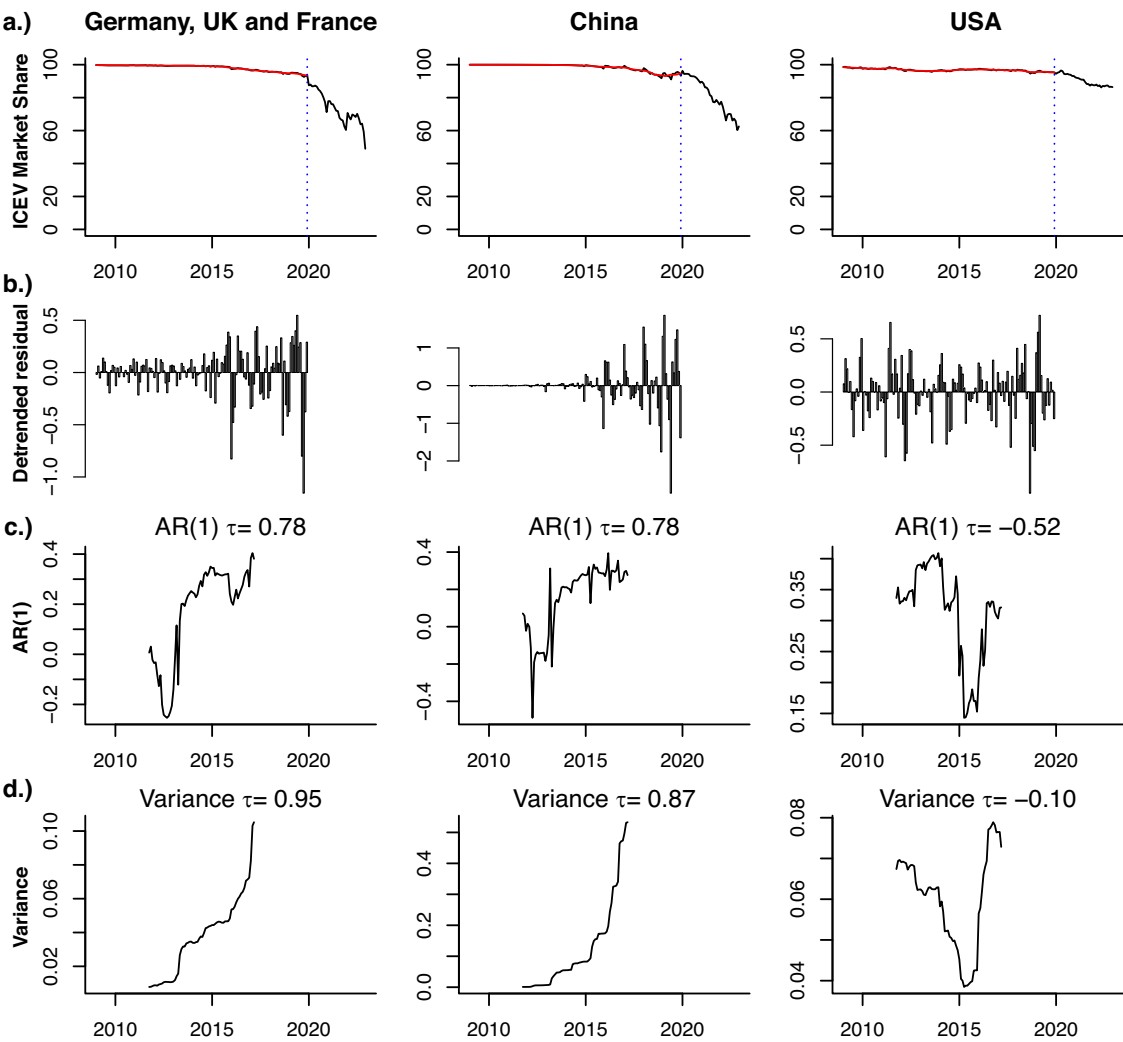

**Fig. 4 | ICEV Market share with loss of resilience indicators. a** Time series of Internal Combustion Engine Vehicle (ICEV) market share for various markets. The blue dashed vertical line denotes the start of 2020, where marked declines in ICEV market shares begin, hence the time series are cut prior to measuring resilience signals. The effect of cutting at 2020 also removes the impact of COVID-19 on sales and resilience. The red line shows the trend line. **b** The detrended series, constructed by subtracting the red trend line from the original time series. Once the sales time series have been cut and detrended, the lag-1 autocorrelation (AR(1)) is calculated on a moving window of 66 points (or equal to half the length of the time series) and plotted in the middle of the window. **c** The AR(1) of the detrended time series. Mann-Kendall tau values above each plot show the tendency of the indicator time series (see Methods). **d** Same as (**c**) but for variance.

transition in countries where agency on vehicle sales and manufacturers is limited.

It is important to note that the reorganisation and retooling of vehicle production lines is costly and involves profit expectations over at least a decade. Therefore, the declining ICEV model variety in the EU and US can be seen as a turning point for the vehicle industry. Meanwhile, we showed evidence that the EV market and the associated charging infrastructure grow as they coevolve. Although we cannot observe the actual intentions of the vehicle industry, once investment is sunk into converting production lines, a return to ICEVs becomes comparatively expensive, irrespective of the oil price. But the exact pace of the transition remains uncertain, as it is sensitive to policy actions, notably the imposition of new tariffs.

A transition to EVs also has many economic advantages that may confer it significant political support reinforcing the identified tipping point. First, it reduces air pollution from fuel combustion and associated morbidity and mortality, via reductions in emissions of particulate matter. Second, it necessarily reduces foreign oil dependence of importing countries, reduces related trade imbalances and increases

international purchasing power. For example, in India, a transition to EVs could save USD75bn/y on oil imports by around 2030 and even up to USD140bn/y by 2040, despite possible increases in vehicle imports[54]. To achieve the EV transition, our analysis suggests that policymakers could facilitate the continued development of increasing varieties and sales volumes of EVs to accelerate cost reductions and match the EV supply to specific mobility preferences in each country. Meanwhile, it may be critical to address the unintended consequences of the EV transition on employment in the traditional vehicle industry, which may have regional economic implications for some socioeconomic groups.

## Methods
### Criteria to identify a tipping point
We use several sources of evidence to identify a tipping point, which in this study we define as a point in time at which a country's vehicle fleet moves towards a state of EV dominance in a self-propelling manner, without further external intervention or changes in conditions or public policy. We use four criteria to establish the existence of a

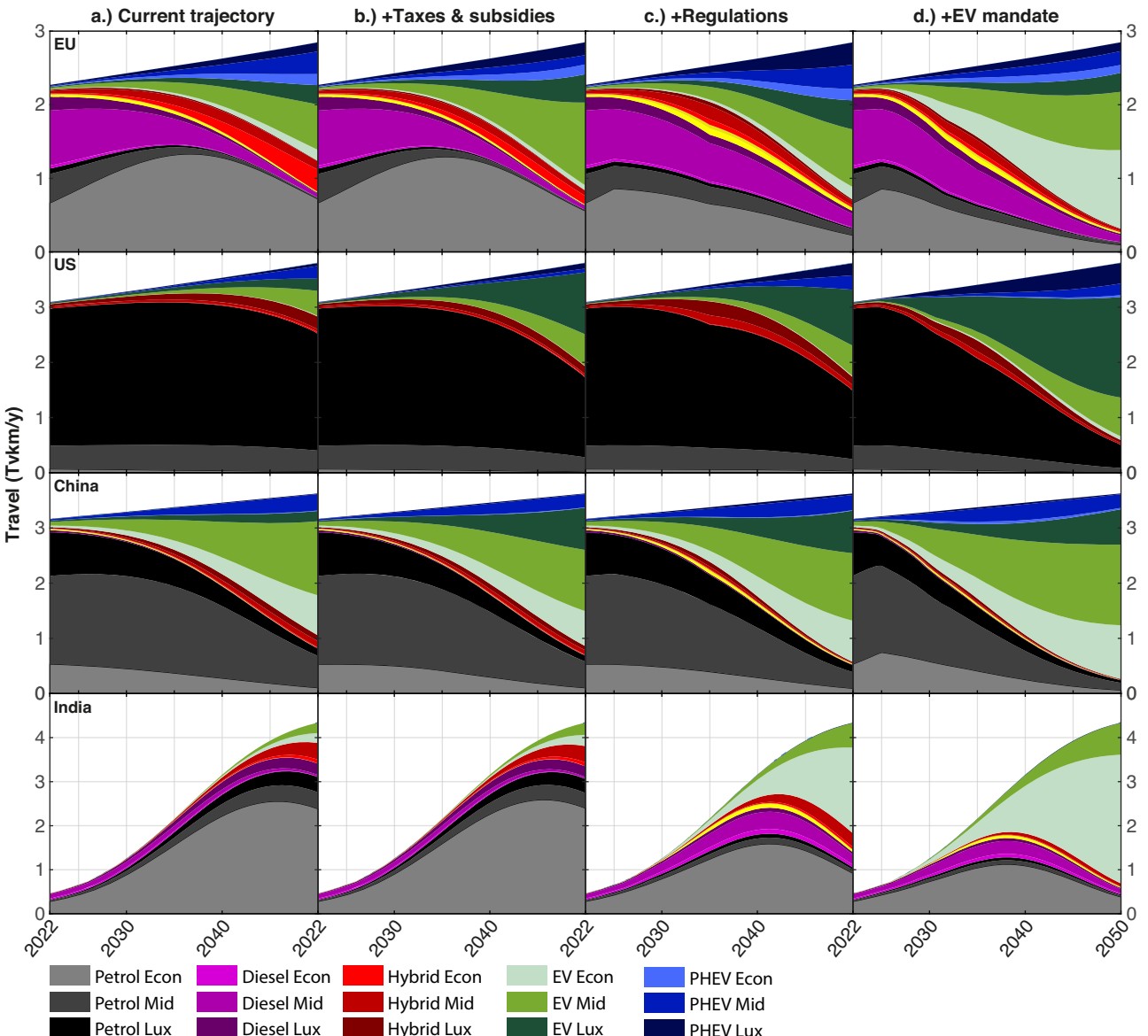

**Fig. 5 | Vehicle technology service generation (in Tera vehicle-km) trajectories under various policy scenarios. a** Technological composition in the current trajectory of fleets (or baseline). **b** Impact of imposing taxes and subsidies on EVs. **c** Impacts of adding phase-out regulations to the taxes and subsidies of column (**b**). **d** Impact of adding EV mandates to the policies in (**c**). Vehicle categories include Petrol, Diesel, Hybrids (combustion), Plug-in Hybrid Electric Vehicle (PHEV), and Electric Vehicle (EV). Vehicle types are represented along three power classes, using the abbreviations Econ Economic, Mid Mid-Range, Lux Luxury (see Methods for details).

tipping point towards EVs: (1) Rapid growth in EV market share and rapid declines in ICEV market share (Figs. 3, 4); (2) an increase in the autocorrelation function in the fluctuations in sales share of ICEVs, indicating loss of resilience of the ICEV system (Fig. 4); (3) the existence of continued cost reductions over time for EVs and of near term points in time of cost parity and price parity between EVs and ICEVs (Supplementary Fig. 7); and (4) transformations in car markets, including changes in the shape of the distribution of sales of ICEVs and EVs, rapid catch-up in EV model diversity compared to ICEVs, and a stop in the growth of ICEV model diversity (Fig. 3 and Supplementary Fig. 6).

Data for the EU, Chinese and US markets all fulfill these criteria except for the loss of resilience indicator in the US, which is inconclusive.

### Vehicle database and sales time series

We obtain or infer EV and ICEV prices and characteristics (power, emissions, fuel use) directly from manufacturer websites and sales numbers from Marklines for each of 32 countries and 329 EV, 222 PHEV and 1901 ICEV models (combined across markets, excluding model variants) between 2016 and 2023, of which 4 regions are shown here. We define three engine/motor power classes following loosely the Eurostat classification: Economic (< 1400cc for ICEVs, <=30 kwh for EVs), Mid-range (>= 1400cc and <2200cc for ICEVs, >30Kwh and <=70 kwh for EVs) and Luxury ( > 2200cc for ICEVs, >70 kwh for EVs). Vehicle prices are roughly lognormally distributed with different means, standard deviations, and medians in each country[29]. We accumulate vehicle data in this way since 2013 and have a yearly time series of price (and other characteristics) distributions starting in 2016 (Supplementary Table 1).

### Vehicle sales

The data for vehicle sales by model and technology (i.e., ICEVs, EVs, PHEVs etc.) were obtained from MarkLines, which provides sales data by brand-model since 2004 for more than 32 countries. To verify that MarkLines comprehensively covers all EV and ICEV models

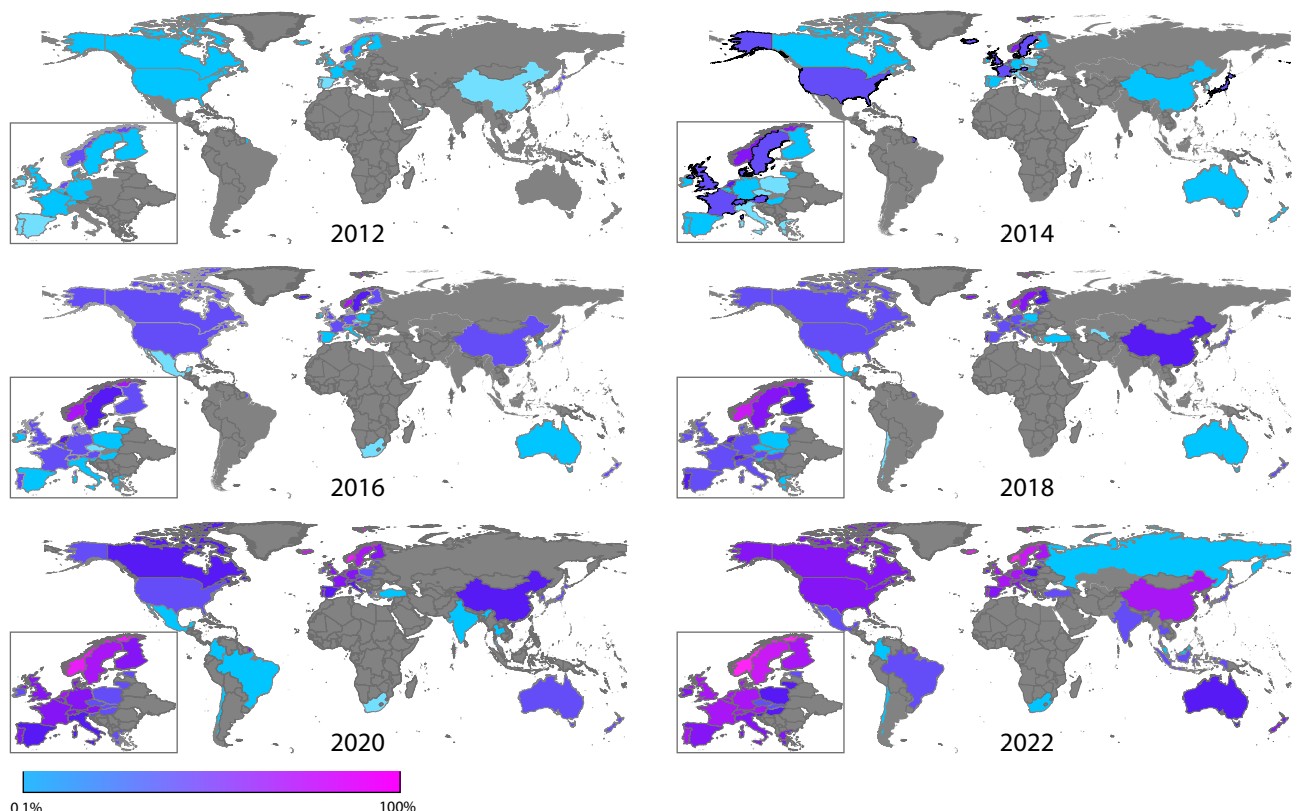

**Fig. 6 | Spillovers of EV deployment from lead markets into peripheral and developing markets from 2012 to 2022.** Colours refer to market shares according to the color bar. The base map image is the intellectual property of Esri and is used herein under license. Copyright © 2025 Esri and its licensors. All rights reserved.

available and sold in each country in a way that is consistent with official national statistics, we carefully check that total MarkLines vehicle sales across models match the annual sales data from official sources (IEA, national yearbooks, Eurostat, US Bureau for Transport Statistics).

**Vehicle prices.** Since vehicle prices can vary significantly between countries for identical brand-models, we visited each manufacturer's website for each brand-model in China, the US, Germany, France, the UK, India, and collected the suggested retail prices (MSRP), engine sizes (for ICEVs, HEVs and PHEVs) or battery sizes (EVs and PHEVs), energy consumption, weight and driving ranges (EVs only). We find that price variations across EU countries are generally within a range of 20%, and therefore we use prices in Germany as a proxy for several EU countries where data is not otherwise available in languages we can interpret. In cases where a brand-model is sold in some EU countries according to Marklines but is not available in Germany, we use price and specifications data from the manufacturers' websites in France or the UK. We further verify the brand-model coverage of MarkLines by cross-comparison with the manufacturer website data thus obtained.

**Measuring resilience loss prior to a tipping point**
As a system loses resilience as it approaches a tipping point, its restorative feedbacks weaken and therefore its ability to return to some equilibrium state declines[55]. This change, known as Critical Slowing Down, can be detected with Early Warning Signals (EWS)[37]. These EWS measure the return rate of a system to equilibrium; a slow return rate indicates a less resilient system, which is closer to a tipping point, than one that has a faster return rate. Figure 1b shows how the potential well of the state the system occupies shallows as the system approaches tipping, representing these restoring feedbacks

weakening. Visually, one can imagine the ball representing the state of the system taking longer to return to the bottom of the well after being perturbed as the system moves towards tipping.

One way to measure changes in resilience is by measuring the lag-1 autocorrelation (AR(1)) of a time series on the system on a moving window, which measures the correlation between the system at time t and t + 1. As the system takes longer to recover, AR(1) will increase as each time step is more correlated to the last, with AR(1) tending towards 1 as a tipping point occurs[56].

Variance is also expected to increase over time[37], as the restoring feedbacks of the incumbent system degrade, the system can sample more of the surrounding state space under the same level of perturbations. For true early warning of the approach towards tipping, one expects both AR(1) and the variance to increase together[57], in cases where the system is forced with identically distributed white noise. In all cases, these indicators are known to pick up general losses in resilience over time in the system, although they may be influenced by intervention[38].

We apply these EWS to measure the loss in resilience of the incumbent regime. For the EV transition the incumbent regime is an ICEV-dominated market, represented here with ICEV market share. This is considered across the national markets in the UK, Germany, France, China, and the US. The ICEV share time series are cut at the start of 2020, prior to the tipping points, and have the trend removed using a kernel regression smoother. On a moving window equal to half the length of the time series, we measure the AR(1) and variance on the resulting residual time series, moving this window one month at a time to get a time series of the indicator series. We then use a Mann-Kendall[58] test to measure the tendency of the indicators, with Kendall's tau = 1 suggesting the indicator is always increasing, -1 always decreasing, and 0 no overall trend, this is approach is common in the literature for assessing these resilience trends.

## Summary of the FTT:transport model

The Future Technology Transformations (FTT) family of models[41,59] (FTT:Power[39], FTT:Transport[40,41], FTT:Heat[60], and FTT:Steel[61]) is a bottom-up representation of technological change that projects future diffusion patterns for individual technologies calibrated on observed trends. The FTT framework models technological diffusion and investment choices using a set of coupled dynamical diffusion and experience curve equations that generate path-dependent S-curve technological change profiles[22,23]. Under the FTT framework, consistent with the sociological empirical literature[3,28], consumers are proportionally more likely to choose a technology that has a higher market share due to its higher availability, visibility, social influence, and due to network effects. This generates the observed early adopter and early majority impacts on the diffusion lifecycle, which is well established empirically[22]. FTT:Transport is fully defined in Supplementary Note 2.

FTT:Transport is parameterised along various dimensions by extracting averages and standard deviations for each vehicle class (Economic, Mid-Range, Luxury) in each technology category (Petrol, Diesel, Hybrid, EV, compressed natural gas, biofuels, hydrogen) from the database. Levelised costs of transportation values, expressing an expected discounted cost per kilometre driven, are estimated at each time step for each vehicle category, including the effects of experience curves and pecuniary policies (existing or assumed subsidies and taxes of various kinds). An evolutionary discrete choice model is used to determine the choice of agents between all vehicle options that are available to them[41,62]. Consumer discount rates of 20% are assumed, equal between all vehicle categories.

The intangibles component of the levelised costs is then estimated as the value that ensures that the modelled diffusion trajectories match the observed diffusion trajectories[63]. The model's baseline is thus by construction designed to extend into the future the current trajectory of diffusion of the various transport technologies in each of the markets modelled. In some regions, ICEVs are already in absolute decline, in others they are not. Global experience curves are used for each technology, and notably for batteries, and for EVs themselves without the battery, assuming that vehicle manufacturers operate and carry knowledge across borders (see Supplementary Note 1).

## Modelling policies in FTT:Transport

*Fuel economy standards* require automakers to design more efficient vehicles, to shift sales toward more efficient models and discontinue inefficient models. Fuel economy regulations are complex as they apply to averages over sales. Given that we don't model sales specifically per manufacturer, we make simplifications. Here, efficiency standards are modelled by adjusting the market shares of the technologies exogenously to meet targets. In the presence of a fuel economy regulation, we assume that there are no new market shares gained in the categories being phased out, leaving the existing vehicles to naturally come to end of their statistical lifetime, leading to exponential declines with half-lives or around 12 years.

*EV mandates* require automakers to sell EVs in numbers exceeding a predefined percentage of their total sales. Here, we set exogenous market share additions at specific points in time to be consistent with the mandate, allowing EV market shares to exceed the mandated value.

*Taxes and subsidies* are modelled through their effect on the generalised Levelised Cost of Transportation (LCOT), calculated as a probability distributed quantity every time step, driving purchasing decisions in the model[40,41]. Pecuniary incentives include fuel taxes, road taxes, ownership taxes and EV/PHEV subsidies. Costs occurring in the future relative to when decisions are taken are assumed discounted by consumers using a consumer discount rate of 20% or less according to scenario choices. Given that consumer discount rates are challenging to measure, we deliberately use a conservatively high value but

vary it as a sensitivity to test our model (lower discount rates bring forward cost and price parity, see Supplementary Fig. 7).

Detailed policy stringencies are given in Supplementary Table 6, with policy sensitivities in Supplementary Tables 16-19.

## Modelling parity in EV prices and total cost of ownership

We construct a relationship between battery prices and EV/PHEV prices according to manufacturing costs and prices observed in the market (Supplementary Figs. 1–3). Electric vehicle manufacturing costs are estimated based on manufacturers' suggested retail prices and a cost-to-price markup factor. Vehicle prices are distinguished from vehicle manufacturing costs due to automaker profits and dealer markups. The markup factors (obtained from ref. 6) vary across vehicle segments, where markup factors for small vehicles tend to be lower than larger ones, but appear to remain stationary over time (see Supplementary Note 1).

Manufacturing costs for EVs depend on four key factors: battery price, platform choice, driving range, and the vehicle's energy use efficiency (where more efficient use of electricity allows for smaller and cheaper batteries for a given driving range). Among these key factors, the fall in the cost of batteries typically accounts for 75% of current falls in EV manufacturing costs[6]. Battery costs are estimated dynamically quarterly with an experience curve as a function of cumulated sales (see below and in Supplementary Note 1). In FTT, changes in the cost of batteries are represented in vehicle manufacturing costs to determine prices on the basis of markups.

The experience curve[23,45] is based on the empirically observed phenomenon that the unit cost of a technology declines by a constant percentage for each doubling of cumulative production volume (e.g., cumulative installed capacity), as described by the equation (1) (Supplementary Note 1):

$$C_i(t) = C_i(t_0) \left( \frac{W_i(t)}{W_i(t_0)} \right)^{-b_i}$$

Where $W_t$ is the total global cumulative production of the technology at time $t$ (i.e., the total kWh capacity of cells produced), $C$ is the cost per unit (USD/kWh), $C_O$ and $W_O$ are reference cost and cumulative production values at the start time $t_0$ of scenarios, while $b_i$ is the experience curve exponent. The latter is related to the cost reduction proportion that results from every doubling of production, what is known as the experience curve rate (ER) in equation (2):

$$ER = 1 - 2^{-b_i}$$

The experience curve rate for EVs in the literature is around $20 \pm 5\%$[7,12,45], while we measure 15-21% with our own data (Supplementary Fig. 1). In the initial stages of diffusion, such as is the case for EVs, costs typically decline rapidly as investment and production scales up exponentially. Our evidence suggests that manufacturing cost reductions have largely been reflected in reductions in EV prices per unit battery capacity (Supplementary Note 1). We find that resources saved via lower battery costs have been re-deployed into a combination of lower EV prices and larger batteries. In later stages of diffusion, such as is the case for ICEVs, doubling the capacity becomes infeasible, and costs do not decline despite that an experience curve rate exists for ICEVs as well.

## Critical minerals analysis

Engineering models exist that compile information relating physical properties of different batteries to allow the estimation of manufacturing costs based on factors including cell chemistry and critical metal prices. Here we use the Battery Performance and Cost (BatPaC) model[64] and the Cell Energy and Cost model (CellEst)[65]. One missing

component in BatPaC are fluctuations over time in the prices of raw metals on the costs of cathode materials. As a supplement to BatPaC, we use CellEst to calculate battery cathode costs based on detailed future metal costs following a scenario approach.

Sensitivity analyses (Supplementary Note 5) include ICEV and EV cost analyses considering various mineral price scenarios and their effect on EV cost parity across six battery chemistries, including variations of Nickel-Manganese-Cobalt (NMC), Lithium-Nickel-Cobalt-Aluminium Oxides (NCA), and Lithium-Iron-Phosphate (LFP), under three price scenarios. Mineral prices were projected based on the methodology found in Zhang et al.[66]. Despite high mineral prices, our findings indicate that EVs in all segments are projected to attain price parity between 2025-2030 even under high mineral price scenarios.

### Charging infrastructure analysis

In all countries where EV deployment has been observed, both the numbers of EVs and of charge points increase approximately exponentially. However, the absolute number of chargers increases with the adoption of EVs (Supplementary Fig. 11), but the number of chargers per EV decreases with the share of EVs in the fleet (Supplementary Fig. 12). Furthermore, the ratio of charging points per EV decreases as the stock share and the proportion of fast chargers increases. We identified a correlation between the amount of EV charging infrastructure per unit of EV stock and the EV stock shares in the fleet in the US, China, EU and rest of the World (RoW) at the region-year level (Supplementary Fig. 12). When the EV share is low, the number of public chargers per EV tends to be high since initial infrastructure development generally exceeds EV sales to support vehicle electrification[18,67]. There is significant variation in residential charging accessibility across cities and countries. For example, fewer households in China have access to private chargers than in the US and Norway where there are a lot more single-unit houses. Charging infrastructure per EV is higher in countries where demand for public chargers is high. On the other hand, in countries such as Norway and the US, fewer public chargers are sufficient for a higher number of EVs, although reliance on public charging solutions increases as EVs are increasingly used for long/interurban journeys[67]. Eventually, as occurred in Norway, once the demand for and the utilization of charging points becomes high enough, the deployment of public charging points becomes increasingly based on commercial decisions where government financial support is no longer needed[68].

Our data suggest that initially deploying a high amount of charging infrastructure per EV in the initial stages of diffusion enables to start the diffusion process, in some cases up to two charge points per EV (Supplementary Figs. 11, 12). In most countries, this was done by the public sector. But this initial infrastructure buildup typically makes a small fraction of the total amount of charging infrastructure deployed in the later stages of diffusion, where typically the number of charging points per EV required to sustain a large population of EVs converges to values around 0.2. Therefore, charging infrastructure can form a barrier for the diffusion of EVs in the very early stages of diffusion, but the evidence does not suggest so for the later stages.

### Data availability

The data and assumptions to operate FTT:Transport are available at https://zenodo.org/uploads/10690730. The data underpinning the analysis is available at https://zenodo.org/records/17508909. Model names are obscured in line with MarkLines data licensing requirements. The full data can be provided to users that hold a MarkLines license upon request by writing to the corresponding author.

### Code availability

The code for FTT:Transport is availble at https://github.com/aileenlam28/FTT-Transport. The code used to study the early tipping signals is available at https://github.com/jbuxt/EV_EWS/tree/main.

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

## Acknowledgements

The authors acknowledge the Global Systems Institute (Exeter) for support. J.F.M. and A.L. acknowledge the World Bank's Climate Support Facility. J.E.B., C.A.B., and T.M.L. acknowledge the Bezos Earth Fund. J.F.M., A.A., and T.M.L. acknowledge the UK Government's BEIS/DESNZ-funded Economics of Energy Innovation and System Transition (EEIST) program (www.eeist.co.uk). The authors thank Simon Sharpe, Femke Nijsse, Somik Lall, Kevin Carey, Etienne Espagne, and Indermit Gill for comments.

## Author contributions

J.F.M. designed and coordinated the research, with support from all authors, designed the FTT model theory and method and wrote the article with support from all authors. A.L. gathered the data, designed and performed the research, built the FTT model, ran the simulations, and co-wrote the text. J.E.B. and C.A.B. applied the early tipping signals method to data. T.M.L. designed the early tipping signals theory and method and co-wrote the text. A.A. supported data collection.

## Competing interests

The authors declare no competing interests.
