## [Transparent Peer Review file · Nature Communications]

Evidence of a cascading positive tipping point towards electric vehicles

Corresponding Author: Professor Jean-Francois Mercure

Version 0:

Reviewer comments:

Reviewer #1

(Remarks to the Author)

This paper scrutinizes temporally when electric vehicle (EV) adoption will become self-propelling. The analysis is executed leveraging data from multiple markets, and surmises that an EV transition is likely in the next few years. The precise timeline varies by market with the United States – a key automarket – experiencing an EV tipping point by 2030 at the latest. Other markets like China and the European Union are expected to experience this tipping point sooner. The authors provide constructive data to support their supposition, the most notable being auto prices, battery prices, and market share. As far as papers go that predict the future state of EV adoption, I find this draft to be a worthy read. That said, I am unable to fully gauge the contribution of the paper to the literature owing to some methodological gaps. I am hopeful that the authors can address these where appropriate.

Author Statement: EVs have seen their costs decline drastically following large scale investment, anticipating or benefiting from favourable policy environments, particularly in the world's leading car markets of Europe, North America and China.

Reviewer Response: Are you referring to the production cost of the vehicle or the MSRP? Generally, EV production costs have decline but it is unclear whether these declines have been met with proportional decreases in MSRP. Do you have data suggesting this is the case on a market-by-market basis? If so, it would be helpful to include that information here.

Author Statement: It could conceivably occur with EVs once ownership costs and the diversity of models available on markets become comparable with incumbent alternatives,

Reviewer Response: This is an important point and worthy of further discussion. There are two separate issues here. The first is the ownership cost and the second is the availability of vehicles that offer comparable performance characteristics as the incumbent. Regarding ownership costs, I would caution the authors that TCO – while popular as a metric in the research community – may not be how consumers think about a product. A new water heater may save consumers thousands every year but it is unlikely to be purchased if the upfront price is too dear for the consumer.

Author Statement: A transition to EVs also has many economic advantages that may confer it significant political support. First, it reduces air pollution and associated morbidity and mortality, via reductions in emissions of particulate matter. Second, it necessarily reduces oil dependence. It can reduce trade imbalances and increase international purchasing power.

Reviewer Response: The authors may want to also offer a different perspective. Regarding reductions in particulate matter, there is also evidence that EV may increase pollution (<https://www.emissionsanalytics.com/news/gaining-traction-losing-tread>). Regarding trade imbalances, I am unsure what precisely the authors are alluding to. Are you suggesting that driving an EV reduces a trade imbalance? If so how. Or are you suggesting that reduced dependence on FOREIGN oil imports bolsters internal security and reduces a trade deficit.

I will now present some methodological issues that warrant further treatment from the authors.

Author Statement: We obtain or infer EV and ICEV prices and characteristics (power, emissions, fuel use) directly from

manufacturer websites and sales numbers from Marklines for each of 32 countries and 329 EV, 222 PHEV and 1901 ICEV models (combined across markets) between 2016 and 2023, of which 4 regions are shown here.

Reviewer Response I am having trouble following the authors' approach here. You state that your model accounts for 329 EV models available for sale between 2016 and 2023. Do you mean models or trims – that is multiple variants of the same model? When assessing price, it would be helpful to know which variant was considered and if you only considered the baseline model, a rationale for why would be timely. More broadly, I think the authors should consider including a list of all the models/trims that were included in the study and stratify it by country (so it is clear which models were available where geographically for sale).

I raise this issue in part because the authors subsequently state “. To verify that MarkLines comprehensively covers all EV and ICEV models available and sold in each country in a way that is consistent with official national statistics, we carefully check that total MarkLines vehicle sales across models match the annual sales data from official sources (IEA, national yearbooks, Eurostat, US Bureau for Transport Statistics).”

A tabulated list of model, year, and country of sale would be helpful in verifying this statement.

Author Statement: Suppl. Figure 2 | Cost and price reductions of EVs in selected models. Time series for 14 vehicle models across the UK (5), US (3), Germany (DE, 3) and France (FR, 3) used to explore the relationship between vehicle prices and battery costs. Data includes vehicle prices, battery costs per kWh and battery capacity.

Reviewer Response: I do not understand why a time series estimate was generated for 14 vehicles versus the 300+ that were considered by the model. It would be helpful to explain this discrepancy. In viewing the time series analysis, battery prices have fallen – however I challenge the assertion that this has been accompanied by a corresponding decrease in price. Consider the following graph in the paper:

Unless I am reading this incorrectly, if we take the US market – the exact price decline appears to be relatively small. The US market does appear to see a drop although the 2023 and 2024 year has largely been characterized by plummeting demand which has forced EV prices to drop. What is clear is that the price declines are more robust in some markets versus others. It would be helpful for the authors to explain why this may be the case.

Author Statement: However, following unprecedented price increase, battery prices have continued to fall in 2023.2 Our findings indicate that EVs in all segments remain projected to attain price parity between 2025-2030 even under high mineral price scenarios

Reviewer Response: The authors may wish to review this paper.

<https://www.nature.com/articles/s41467-024-51152-9>

Author Statement On the basis that EVs generally cost more upfront but less over the lifecycle than ICEVs, price parity can be expected to occur later than TCO parity.

Reviewer Response: From a public policy perspective, this is one of the most important takeaway messages. The reason why is because it impacts a) when governments can reach their decarbonization targets (and that impacts NDCs), and b) how long governments may be required to subsidize EV purchases for (which impacts the national debt). I encourage the authors to consider mentioning/discussing this issue briefly.

(Remarks on code availability)

.

Reviewer #2

(Remarks to the Author)
See the attachment

(Remarks on code availability)

Reviewer #3

(Remarks to the Author)
Review for the paper „Evidence of a cascading positive tipping point towards electric vehicles”

The authors present a simulation model of market diffusion of electric vehicles and tipping-point behavior with endogenous experience curves for EVs and their components and different car classes. They use car sales data for a decent number of

vehicle models and countries to fit and calibrate the experience curves and the long-run simulation model. They provide an interesting empirical analysis and evidence of early signals for approaching a tipping point from ICEVs to EVs in Europe and China, but not yet in the U.S. and India. Subsequently, the authors employ an elaborate technology and market diffusion model to simulate how and when parity on total ownership costs between ICEVs and EVs is reached and self-accelerating tipping to-ward EVs occurs in the three “lead markets” and India as a peripheral market. Finally, they examine the role of transport policies (taxes, subsidies, standards, ...), roll-out of charging infrastructure, and market developments for critical minerals required for EV and battery production.

The analysis is rich, comprehensive, and appears well executed and described. The data basis for the empirical calibrations is good. Methodological choices and execution are overall convincing. I like the level of detail behind the various components. Overall, the paper seems fairly mature. But I would like to point out several comments.

Comments:

1. How do the approach and the results of the paper relate to previous results in the literature? This would be helpful to see more clearly the contribution of the present paper. I understand the application of the empirical method to identify tipping signals to the EV market is original (correct?). But how do the scenario simulations for EV diffusion and tipping compare to existing work on EV market diffusion? My understanding is also that the FTT:Transport technology evolution model which the authors use was preexisting for previous work and is applied to the context of EV market diffusion. So it appears the contribution of the paper does not lie in developing the modelling approach. Are the results with this approach much different from other, possibly simpler, approaches out there (e.g. a more stylized learning curve and accelerating EV adoption once total ownership cost falls below the one of ICEVs)? Does the application to these vehicle sales data let the model/paper stand out or has that also been used before by other work in the context of EV market diffusion? To read and see these advances over possibly less elaborate approaches would help to better see the value in this approach.

2. Analyzing policies such as taxes, subsidies, standards and EV mandates and their potential for accelerating EV adoption and tipping as the authors do is important and valuable. But it is unclear which policy levels they apply in their scenarios and what the analysis implies for policy choice and design. Certainly, the pace of EV adoption and tipping will depend on (carbon, gasoline) tax levels and on subsidy levels, targets for fuel economy regulation. The same holds for EV mandates. Which policy levels are applied? Do they differ between the different countries? How sensitive are the results to different policy levels? Similarly, the scenarios for cost parity in India (Ext. Fig. 6) will depend on policies in India and the other countries. How did they choose policy levels for the scenarios? After what seems most likely? Current policies times a constant factor? Given that the authors are emphasizing the insights on timing of the transition, accounting for these determinants seems first order.

3. Building on Comment 2: When discussing policy levels in the context of EV diffusion as a way of decarbonizing private vehicle transport, the question arises which policy levels should be implemented? Which carbon abatement costs do the policy trajectories discussed in Fig. 5 and the rest of the paper imply? High market penetration of EVs sounds desirable but could be achieved faster (high abatement costs) or more slowly (lower abatement costs) with different policy paths. Since policies are discussed as being complements, how much do combinations of taxes/subsidies and regulations and mandates reduce the implied abatement cost? Is adding EV mandates to existing taxes/subsidies more advisable than increasing the tax by xyz? The authors present a lot of results and adding a full welfare analysis would probably go too far. But implied abatement costs should be added and the trade-offs between different policy instruments in this framework should be discussed. This would provide much additional value to the policy discussion. It would help the reader a lot to better understand what we learn from the paper about a (more) welfare optimal policy mix.

4. Currently many new trade restrictions and tariffs are being and have been introduced (US-China, EU-China, etc.), and it appears as if the world are enter-ing a period in which trade barriers will rather be the rule. How sensitive are the results in the paper to current new and future trade restrictions? A discus-sion with some quantified illustrations of this dimension would be important for the policy message.

5. Typo: There is a spare “policy” in the caption of Fig. 5.

Overall, I would like to recommend revisions and resubmission of the paper.

(Remarks on code availability)

Yes, there is a Readme file with instructions. The code looks reasonably structured and commented. I did not install and run as I do not have a Matlab license right now.

Version 1:

Reviewer comments:

Reviewer #2

(Remarks to the Author)

I have reviewed the response of the authors to my set of concerns and recommendations. I believe that they have responded in a satisfactory way. While we may have a degree of difference regarding the endogeneity of certain observable conditions - i.e., movement along the experience curve, cost-cutting technological innovations, networking scale economies, etc. may have been the causal consequence of a single regulation, i.e., the phase out of ICEVs by 2035 the response of the authors is reasonable.

I imagine that the manuscript will make a contribution to the relevant literature.

(Remarks on code availability)

Reviewer #3

(Remarks to the Author)

The updates and changes in the paper answer all my previous comments sufficiently.

(Remarks on code availability)

REVIEWER COMMENTS

Reviewer #1 (Remarks to the Author):

This paper scrutinizes temporally when electric vehicle (EV) adoption will become self-propelling. The analysis is executed leveraging data from multiple markets, and surmises that an EV transition is likely in the next few years. The precise timeline varies by market with the United States – a key automarket – experiencing an EV tipping point by 2030 at the latest. Other markets like China and the European Union are expected to experience this tipping point sooner. The authors provide constructive data to support their supposition, the most notable being auto prices, battery prices, and market share. As far as papers go that predict the future state of EV adoption, I find this draft to be a worthy read. That said, I am unable to fully gauge the contribution of the paper to the literature owing to some methodological gaps. I am hopeful that the authors can address these where appropriate.

Author Statement: EVs have seen their costs decline drastically following large scale investment, anticipating or benefiting from favourable policy environments, particularly in the world's leading car markets of Europe, North America and China.

Reviewer Response: Are you referring to the production cost of the vehicle or the MSRP? Generally, EV production costs have decline but it is unclear whether these declines have been met with proportional decreases in MSRP. Do you have data suggesting this is the case on a market-by-market basis? If so, it would be helpful to include that information here.

Many thanks for the above comments and for this clarification request. We have information on both vehicle costs and prices. We gather MSRPs in each region (Fig 3), while we estimate costs using battery unit costs as proxies using published values (Fig 2 combined with battery capacity values, Suppl. Fig 2-3)) For other manufacturing costs, we don't have detailed data. Suppl. Note 1 explains our methodology, and Suppl. Fig 2 shows that within some data variations and uncertainty, declines in MSRPs are roughly proportional to declines in battery costs. Suppl. Fig. 2-3 together show that manufacturing cost reductions over time have been re-deployed over time as a combination of greater battery capacities and lower MSRPs in the three leading markets.

Author Statement: It could conceivably occur with EVs once ownership costs and the diversity of models available on markets become comparable with incumbent alternatives,

Reviewer Response: This is an important point and worthy of further discussion. There are two separate issues here. The first is the ownership cost and the second is the availability of vehicles that offer comparable performance characteristics as the incumbent. Regarding ownership costs, I would caution the authors that TCO – while popular as a metric in the research community – may not be how consumers think about a product. A new water heater may save consumers thousands every year but it is unlikely to be purchased if the upfront price is too dear for the consumer.

We agree with the reviewer. There are two points here, (1) comparable performance and characteristics across a wide range of vehicles, and (2) TCO as a metric.

(1) We cannot enumerate all the characteristics that each vehicle in the dataset have and compare that against all EVs in the dataset. However, we can use as a proxy for model variety the price distribution, which to some extent represents this (Figure 3 and Ext Fig. 2). We can see clearly that EV and PHEV variety started very low and increased explosively over time, particularly in China, to gradually come to match or nearly match the variety of conventional vehicles. We explain this in Suppl. Note 6, and Suppl. Figure 7 illustrates this. We consider reaching variety parity a critical milestone in the transition, at least as important as reaching TCO parity.

(2) For modelling in FTT, we estimate three TCO values by splitting the price distributions into three ranges according to vehicle power: the economic, mid-range and luxury ranges. This is how we represent model variety as well as cost declines. TCO parity is one amongst several criteria we use to identify the tipping point, others including model variety (Fig. 3 and Ext. Fig. 2), the market share growth (Ext. Fig. 4) and the autocorrelation function to detect critical slowing down (Fig. 4). Thus we are not entirely reliant on TCO. It is true that agents may not all think in terms of TCO, which we define as the levelized cost, but, given that it involves costs over time and a discount rate, it is comparable to what agents may pay monthly for a lease contract for a vehicle + fuel, therefore it cannot be too far from how agents perceive costs (assuming they have access to financing).

Author Statement: A transition to EVs also has many economic advantages that may confer it significant political support. First, it reduces air pollution and associated morbidity and mortality, via reductions in emissions of particulate matter. Second, it necessarily reduces oil dependence. It can reduce trade imbalances and increase international purchasing power.

Reviewer Response: The authors may want to also offer a different perspective. Regarding reductions in particulate matter, there is also evidence that EV may increase pollution (<https://www.emissionsanalytics.com/news/gaining-traction-losing-tread>). Regarding trade imbalances, I am unsure what precisely the authors are alluding to. Are you suggesting that driving an EV reduces a trade imbalance? If so how. Or are you suggesting that reduced dependence on FOREIGN oil imports bolsters internal security and reduces a trade deficit.

Many thanks for these comments, we have amended these two sentences to say “A transition to EVs also has many economic advantages that may confer it significant political support. First, it reduces air pollution **from fuel combustion** and associated morbidity and mortality, via reductions in emissions of particulate matter. Second, it necessarily reduces **foreign** oil dependence **of importing countries, and** reduce **related** trade imbalances and increase international purchasing power.”

Hence we focus the statement on pollution from fuel combustion only (the above reference the reviewer kindly suggests, which we acknowledge, does not provide estimates of pollution increases from tyres due to EVs being heavier, to compare against fuel combustion).

The statement on trade imbalances, explained in detail in ref. 53, originates from the relative impact on the trade balance between avoided oil and importing EVs (where relevant). Once past total ownership cost parity, importing EVs turns the balance in favour of reducing trade imbalances due the value of avoided oil starting to exceed

the difference in value of between EVs and ICEVs (given electricity is overwhelmingly a domestic product).

I will now present some methodological issues that warrant further treatment from the authors.

Author Statement: We obtain or infer EV and ICEV prices and characteristics (power, emissions, fuel use) directly from manufacturer websites and sales numbers from Marklines for each of 32 countries and 329 EV, 222 PHEV and 1901 ICEV models (combined across markets) between 2016 and 2023, of which 4 regions are shown here.

Reviewer Response I am having trouble following the authors' approach here. You state that your model accounts for 329 EV models available for sale between 2016 and 2023. Do you mean models or trims – that is multiple variants of the same model? When assessing price, it would be helpful to know which variant was considered and if you only considered the baseline model, a rationale for why would be timely. More broadly, I think the authors should consider including a list of all the models/trims that were included in the study and stratify it by country (so it is clear which models were available where geographically for sale).

All data points represent independent vehicle models, not variants (we picked a midpoint variant per model using our judgment). We amended the first sentence in the Methods section: “We obtain or infer EV and ICEV prices and characteristics (power, emissions, fuel use) directly from manufacturer websites and sales numbers from Marklines for each of 32 countries and 329 EV, 222 PHEV and 1901 ICEV models (combined across markets, **excluding model variants**) between 2016 and 2023, of which 4 regions are shown here.” We picked the median model variant where several were available at different price ranges for different characteristics. We comment on releasing data below.

I raise this issue in part because the authors subsequently state “. To verify that MarkLines comprehensively covers all EV and ICEV models available and sold in each country in a way that is consistent with official national statistics, we carefully check that total MarkLines vehicle sales across models match the annual sales data from official sources (IEA, national yearbooks, Eurostat, US Bureau for Transport Statistics).”

A tabulated list of model, year, and country of sale would be helpful in verifying this statement.

Given the lack of comprehensive public data on vehicles, we constructed our dataset using the subscription-only dataset from Marklines on vehicle sales by model, which we matched manually to vehicle characteristics obtained from manufacturer websites. The downsides of using a commercial dataset include strict licensing rules regarding re-distribution: we are not entitled to redistribute raw Marklines data to non-subscribers or publicly (we can re-distribute the rest, which we gathered on dealer websites). Hence we are contractually obliged to remove vehicle model names from any data we release, so long as it includes sales numbers. We nonetheless consider our dataset one of the most comprehensive available, even without model names (year, sales numbers, country of sale and prices are available).

Our data (without model names) is downloadable at <https://zenodo.org/records/14295570> as indicated in our data availability statement, where we added “Model names are obscured in line with MarkLines data licensing requirements” for clarity. We hope the reviewer agrees this is the best we can do.

Author Statement: Suppl. Figure 2 | Cost and price reductions of EVs in selected models. Time series for 14 vehicle models across the UK (5), US (3), Germany (DE, 3) and France (FR, 3) used to explore the relationship between vehicle prices and battery costs. Data includes vehicle prices, battery costs per kWh and battery capacity.

Reviewer Response: I do not understand why a time series estimate was generate for 14 vehicles versus the 300+ that were considered by the model. It would be helpful to explain this discrepancy. In viewing the time series analysis, battery prices have fallen – however I challenge the assertion that this has been accompanied by a corresponding decrease in price. Consider the following graph in the paper:

Establishing a relationship between cost and price over time requires uninterrupted timeseries for specific vehicle models. Vehicle models are rapidly introduced and discontinued from markets, which restricts how many timeseries we can find. We do not have time series for battery capacities for various models in the database, due to data gaps, further restricting the number of full length timeseries available. What remains as complete timeseries are the 14 models we show in Suppl. Fig. 2-3, covering three regions. The main text of Supplementary Note 1 indicates this: “For other vehicle models, battery capacity values in our dataset do not cover the entire historical period.”

Unless I am reading this incorrectly, if we take the US market – the exact price decline appears to be relatively small. The US market does appear to see a drop although the 2023 and 2024 year has largely be characterized by plummeting demand which has forced EV prices to drop. What is clear is that the price declines are more robust in some markets versus others. It would be helpful for the authors to explain why this may be the case.

We acknowledge this comment from the reviewer. We point out that pricing is a manufacturer and dealer choice, which does not necessarily have a simple explanation that can be evidenced or modelled. Thus our data allows us to establish the following (quoting Suppl. Note 1):

“In Suppl. Figure 3, we show that vehicle prices scale with battery costs by a factor 5 in Europe (UK, Germany, France) and 2 in the US. This means that, not only are battery cost reductions passed on to consumers, but cost reductions achieved in the manufacturing of the rest of the vehicles are also passed on to consumers, at a similar rate (our data does not allow to distinguish the two rates). This supports our assumption that learning cost reductions in the manufacturing of EVs are reflected in vehicle prices.”

We are not aware of evidence documenting any significant movements in EV prices to be related with stagnating demand (we do not observe any relationship between pricing and demand). As far as the evidence that we have seen goes, price declines originate from lower manufacturing costs (Fig 2), while price increase are related with material cost increases (according to BNEF, see ref 2).

Author Statement: However, following unprecedented price increase, battery prices have continued to fall in 2023.² Our findings indicate that EVs in all segments remain projected to attain price parity between 2025-2030 even under high mineral price scenarios

Reviewer Response: The authors may wish to review this paper.

<https://www.nature.com/articles/s41467-024-51152-9>

Many thanks for the above reference, which we now include at the end of the results section (paragraph before last). That paper, however, only deals with US sources of materials for US sales of EVs, and does not provide estimates of material cost/price changes in its scenarios. Material scarcity can be supplied both domestically and via imports. Understanding price dynamics requires a global analysis. For that, we use instead the approach given the Methods section 'Critical minerals analysis'.

Author Statement On the basis that EVs generally cost more upfront but less over the lifecycle than ICEVs, price parity can be expected to occur later than TCO parity.

Reviewer Response: From a public policy perspective, this is one of the most important takeaway messages. The reason why is because it impacts a) when governments can reach their decarbonization targets (and that impacts NDCs), and b) how long governments may be required to subsidize EV purchases for (which impacts the national debt). I encourage the authors to consider mentioning/discussing this issue briefly.

We agree with this statement. Given tight space constraints, we added to the 10th paragraph of the Results section "Our projections of cost/price parity (Ext. Fig. 3) can help determine when EV subsidies can safely be discontinued."

Reviewer # 2

Evidence of a cascading positive tipping point towards electric vehicles Referee Report

The manuscript centers on the determination of a "tipping point" for the adoption of EVs in specific lead country markets and globally. To test the relevant hypotheses, the authors use a comprehensive dataset over the composition of car sales between 2016 and 2023, tracking 2,452 models in 33 countries. The authors find that several European markets and China have tipped to EVs, although this is not the case for the US. EV sales are rising exponentially while those of ICEVs are declining. The data and methods of the manuscript predict that the tipping point likely lies within the next few years in lead markets of the EU and China, and potentially the US; thereafter, the tipping point could spill out into peripheral vehicle markets across the rest of the world.

As a measure of the loss of resilience of the incumbent technology, the authors find that the variance and lag-1 autocorrelation of fluctuations in ICEV sales increased beforehand, consistent with the approach to a tipping point. For the purpose of identifying the conditions that can accelerate the convergence to tipping points, the paper uses simulations of technology evolution to identify timescales for cost-parity and policy frameworks that could accelerate the transition to largely eliminate ICEVs before 2050.

The paper is well motivated and generally clear. Two revisions would strengthen the presentation and research project.

We thank the reviewer for taking the time to read our manuscript thoroughly and giving it some thought.

The description of the method is unnecessarily disparate and less integrated than possible. In either the Introduction of Content and Theory section, the authors should organize a single paragraph describing the sequential steps used to identify a “tipping point.” Seemingly, each step corresponds to a section/figure in the Appendix that can be referenced. To the extent that each step/stage functions sequentially or in parallel interactively, it may also help to create a diagram that exhibits the transition from policy to tipping point to EV dominance.

Many thanks, we agree with this. It is difficult at this stage to add much text in the main text without loss of content, therefore to accommodate this, we added to the end of the ‘Context and Theory’ section the sentence “ The methods section describes the steps we use to identify a tipping point.”, in order to put the requested information at the start of the Methods, as follows:

We added to the start of the Methods the section:

“Criteria to identify a tipping point

We use several sources of evidence to identify a tipping point, which in this study we define as a point in time at which a country’s vehicle fleet moves towards a state of EV dominance in a self-propelling manner, without further external intervention or changes in conditions or public policy. We use three criteria to establish the existence of a tipping point towards EVs:

- 1- Rapid growth in EV market share and rapid declines in ICEV market share (Fig. 3-4),
- 2- An increase in the autocorrelation function in the fluctuations in sales share of ICEVs, indicating loss of resilience of the ICEV system (Fig. 4),
- 3- The existence of continued cost reductions over time for EVs and of near term points in time of cost parity and price parity between EVs and ICEVs (Ext. Fig. 3),
- 4- Transformations in car markets, including changes in the shape of the distribution of sales of ICEVs and EVs, rapid catch-up in EV model diversity compared to ICEVs, and a stop in the growth of ICEV model diversity (Fig. 3 and Ext. Fig. 2).

Data for the EU, Chinese and US markets all fulfill these criteria except for the loss of resilience indicator in the US, which is inconclusive.”

We hope that this addition brings clarity and addresses the comment.

The second point is more fundamental; it concerns the basic driver of the transition. Succinctly put, arguably the convergence of market phenomena – the experience curve, cost-cutting technological innovations, networking scale economies, etc. – have all been the causal consequence of a single regulation, i.e., the phase out of ICEVs by 2035.

L 33,34: Plug-in hybrid electric vehicles (PHEVs) overcome issues of range anxiety perceived with EVs , but they are now to be phased out along with ICEVs in the EU and the UK by 2035,...

The zero quota restriction has functioned as an exogenous shock, requiring the conditions for the endogenous evolution of the “tipping point.” The zero quota alone stands as the sufficient condition for the market conditions that have ensued. The subsidies make the time-limited transition more feasible and politically palatable. The concern is that the presentation gives too much weight to the independent force of the markets, while in fact, the behavior of the markets has been shaped by a drop-dead deadline that has made a “tipping point” and the market adjustments inevitable. The story is about how markets adapt to a zero quota over a fixed time horizon, a process that involves a “tipping point.” But this then raises the issue of the precise meaning of “tipping point” and the 7 references to a “self-propelling” transition.

We acknowledge the value in this comment. However, our reply is that it does not appear consistent with the evidence that we show. We are not aware of established evidence on the effectiveness of the announced 2035 bans on conventional vehicles. It is interesting to note that substantial lobbying activity is being done by car manufacturers in the EU requesting governments to repeal those laws. It is not implausible that those laws could be repealed if it becomes unlikely that the EV transition is sufficiently on track for the bans to be fully achieved. The world has also observed drastic policy reversals in the US recently. And lastly, the bans have no immediate impact on the economics of EVs in the near-term (it is not clear what happens if the ban is not achieved).

However, clear evidence exists on the effectiveness of nearer term sales mandates imposed to manufacturers (California and China specifically). These policies have worked largely because they actually bite in the near term with severe penalties, and manufacturers have largely respected them. Thus we find it unlikely that a zero quota has the power to cause a tipping point on its own, while our evidence suggests that cost reductions, model variety, mandates and EV subsidies play a bigger role.

L. 39,40: Hence it is important and timely to examine whether the observed diffusion of EVs could become self-propelling and tip towards a state of dominance.

The meaning of and conditions for the transition to be “self-propelling” need clarification. Does the term mean that the 2035 zero quota and subsidies could be eliminated, still resulting in the convergence to 100% EVs, or does it mean that given the existing body of government policy and subsidies, the adjustment process requires no further government initiative, i.e., the remainder of the transition can be achieved through the fixed set of public-policy conditions prices and incentives. Status quo vs. full market transition with neutral government policy is a critical distinction that is made ambiguous by the term “self-propelling”.

Many thanks, we agree this was not as clear as it could be. With our addition of ‘Criteria to identify a tipping point’ in the Methods, we included a definition for a tipping point: “[...] which in this study we define as a point in time at which a country’s vehicle fleet moves towards a state of EV dominance in a self-propelling manner, without further external intervention or changes in conditions or public policy.”

It does not mean that policy reversals cannot stop a tipping point, they could. It means that without changes in conditions, the state of EV dominance would gradually be achieved in an accelerating manner (‘self-propelling’ causing the acceleration). We also note that it is not clear to us that removing the ban on conventional vehicles would necessarily stop the tipping point from occurring, as per our earlier comment, given such an objective is far in the future and can be repealed.

It appears that the authors have the data and methods to determine or offer informed speculation regarding the two alternative tipping-point and self-promotion scenarios. Even if the scenario is that of status quo policy-driven transition in the EU and China, the spillover effects to the rest of the global market (now including the US likely w/o such direct incentives) is a critical feature of the research.

Many thanks, indeed. Putting aside the thorny problem of tariffs (addressed further down), our data points to the transition rapidly spilling out from lead markets to peripheral markets (Fig. 6). Battery cost reductions also spread internationally via trade. However, this could change with the new tariff situation, which is difficult to analyse because it changes weekly. See our comment on tariffs in our response to reviewer 3 below.

Reviewer #3 (Remarks to the Author):

Review for the paper „Evidence of a cascading positive tipping point towards electric vehicles”

The authors present a simulation model of market diffusion of electric vehicles and tipping-point behavior with endogenous experience curves for EVs and their components and different car classes. They use car sales data for a decent number of vehicle models and countries to fit and calibrate the experience curves and the long-run simulation model. They provide an interesting empirical analysis and evidence of early signals for approaching a tipping point from ICEVs to EVs in Europe and China,

but not yet in the U.S. and India. Subsequently, the authors employ an elaborate technology and market diffusion model to simulate how and when parity on total ownership costs between ICEVs and EVs is reached and self-accelerating tipping toward EVs occurs in the three “lead markets” and India as a peripheral market. Finally, they examine the role of transport policies (taxes, subsidies, standards, ...), roll-out of charging infrastructure, and market developments for critical minerals required for EV and battery production.

The analysis is rich, comprehensive, and appears well executed and described. The data basis for the empirical calibrations is good. Methodological choices and execution are overall convincing. I like the level of detail behind the various components. Overall, the paper seems fairly mature. But I would like to point out several comments.

Many thanks!

Comments:

1. How do the approach and the results of the paper relate to previous results in the literature? This would be helpful to see more clearly the contribution of the present paper. I understand the application of the empirical method to identify tipping signals to the EV market is original (correct?). But how do the scenario simulations for EV diffusion and tipping compare to existing work on EV market diffusion? My understanding is also that the FTT:Transport technology evolution model which the authors use was preexisting for previous work and is applied to the context of EV market diffusion. So it appears the contribution of the paper does not lie in developing the modelling approach. Are the results with this approach much different from other, possibly simpler, approaches out there (e.g. a more stylized learning curve and accelerating EV adoption once total ownership cost falls below the one of ICEVs)? Does the application to these vehicle sales data let the model/paper stand out or has that also been used before by other work in the context of EV market diffusion? To read and see these advances over possibly less elaborate approaches would help to better see the value in this approach.

Many thanks, indeed, legitimate point to make. We added to the end of the introduction:

“The novelty of this paper is multiple: (1) it assembles a comprehensive dataset 2016-2023 that tracks the evolution of leading vehicle markets; (2) it applies the early warning tipping point signals method to vehicle data; (3) it calibrates FTT:Transport to the most recent data timeseries to make predictions of cost parity and policy effectiveness; (4) it establishes empirical relationships between EV costs, prices and deployment.”

The novelty also comes in assembling these different types of evidence together. The FTT:Transport model is well established and has been widely published before using older datasets. The only comparable alternative to FTT:Transport in terms of

method, breadth and underlying recent dataset is BNEF's EVO model, but it is not published in sufficient detail to assess scientifically, thus we are the first to claim EV tipping points empirically with a scientifically developed method.

2. Analyzing policies such as taxes, subsidies, standards and EV mandates and their potential for accelerating EV adoption and tipping as the authors do is important and valuable. But it is unclear which policy levels they apply in their scenarios and what the analysis implies for policy choice and design. Certainly, the pace of EV adoption and tipping will depend on (carbon, gasoline) tax levels and on subsidy levels, targets for fuel economy regulation. The same holds for EV mandates. Which policy levels are applied? Do they differ between the different countries? How sensitive are the results to different policy levels? Similarly, the scenarios for cost parity in India (Ext. Fig. 6) will depend on policies in India and the other countries. How did they choose policy levels for the scenarios? After what seems most likely? Current policies times a constant factor? Given that the authors are emphasizing the insights on timing of the transition, accounting for these determinants seems first order.

We agree. For that reason, we included a new table with policy stringency details (Suppl. Table 14). We created a range of new sensitivities on policy levels, of which the outcomes are given in Suppl. Tables 15-18. It is important to note that a vast number of possible policy formulations can achieve the same results, including within a given design structure with changes in stringencies. This means in principle that the number of sensitivities to explore is unbounded, and thus we are not able to claim to be comprehensive. We consider that the structure of policy design matters more, when explaining the narrative of this paper, than the exact stringency of individual instruments (e.g. the exact value of a subsidy against the stringency of a mandate when both are used), hence the sensitivity analysis is designed to show that the model outcomes do not strongly depend on those exact numbers. A link to this was added in the Methods and explanations in Suppl. Note 5.

3. Building on Comment 2: When discussing policy levels in the context of EV diffusion as a way of decarbonizing private vehicle transport, the question arises which policy levels should be implemented? Which carbon abatement costs do the policy trajectories discussed in Fig. 5 and the rest of the paper imply? High market penetration of EVs sounds desirable but could be achieved faster (high abatement costs) or more slowly (lower abatement costs) with different policy paths. Since policies are discussed as being complements, how much do combinations of taxes/subsidies and regulations and mandates reduce the implied abatement cost? Is adding EV mandates to existing taxes/subsidies more advisable than increasing the tax by xyz? The authors present a lot of results and adding a full welfare analysis would probably go too far. But implied abatement costs should be added and the trade-offs between different policy instruments in this framework should be discussed. This would provide much additional value to the policy discussion. It would help the reader a lot to better understand what we learn from the paper about a (more) welfare optimal policy mix.

We understand the view of the reviewer, and we've been asked this at various times over the course of our development of the FTT model. In 2022 we published a paper (using FTT:Transport with an older dataset) that looks at the synergies between policy instruments, using hundreds of plausible policy combinations (the full space to explore being unbounded), and determined their cost effectiveness, in <https://doi.org/10.1016/j.erss.2021.101951>, Citation no 37.

We cite this in the section of the main text discussing the policy analysis, and we reproduce its main figure in Suppl. Fig. 8, showing which instrument pairs synergise with one another. It would be beyond the scope of this paper to repeat this exercise with the new dataset, as the outcomes on the direction or magnitudes of synergies are not new and will not change with the new data.

However, to address the question on abatement costs, we added in Suppl. Figure 9 the abatement costs implied in scenario (d) of Figure 5. We discuss this in the new Supplementary Note 8. We see cost parity and negative abatement costs occurring from 2025-2027 onwards, consistent with the tipping point and cost parity narrative. However, those differences lessen over time as a subset of consumers begin to buy *more expensive EVs* with their income, pushing the average up (but keeping in mind that variety is rapidly increasing, implying that other consumers also begin to buy *cheaper EVs* – the distribution is becoming wider). Abatement costs remain negative throughout the period. We included a link to Suppl. Note 8 in the conclusion of the main paper.

4. Currently many new trade restrictions and tariffs are being and have been introduced (US-China, EU-China, etc.), and it appears as if the world are entering a period in which trade barriers will rather be the rule. How sensitive are the results in the paper to current new and future trade restrictions? A discussion with some quantified illustrations of this dimension would be important for the policy message.

We agree with the reviewer, but also point out how thorny this issue is, given that the tariffs now change on a monthly basis. This suggests that doing sensitivities may be more informative than detailed analyses of specific tariffs. The situation is also marred by the fact that vehicles are not always manufactured in the manufacturers' headquarter countries (e.g. Nissans made in the UK), or vehicles assembled in a particular country out of components manufactured elsewhere, in order to avoid tariffs (which happens in Brazil). We do not have at hand a comprehensive database on tariffs imposed on vehicles, nor details on the location of vehicle value chains.

We sharpen the reviewer's question by interpreting it as "could EV tipping points be reversed by the imposition of tariffs on EV or battery imports in the US, EU and/or China?". We focus on these three countries since they are currently the main EV producers and therefore the countries with incentives to protect their EV industry.

We point out the fact that all three producers have the capacity to rely solely on their domestic production for at least a portion of their domestic EV sales, which in principle means that they can all achieve tipping points domestically eventually, although tariffs could cause delays by restricting supply and/or increasing prices. However, there is the important caveat that while the EU and US make batteries, they import a majority from China. We assume that China's lack of reliance on foreign-made components implies that its domestic EV tipping point is unlikely to be affected by tariffs.

For the sake of simplicity, given that all three markets can make all components of cars at scale except batteries, we explore the impacts of tariffs of 50% and 100% imposed on batteries manufactured abroad, for the EU and US only. The results are shown in Suppl. Figure 10, with discussion in the new Supplementary Note 8. We find that while tariffs could cause delays in the diffusion of EVs in the EU and US, tariffs do not invalidate our results regarding the occurrence of tipping points. We included a link to Suppl. Note 8 in the conclusion of the main paper.

5. Typo: There is a spare “policy” in the caption of Fig. 5. Many thanks.

Overall, I would like to recommend revisions and resubmission of the paper.

Many thanks!

Reviewer #3 (Remarks on code availability):

Yes, there is a Readme file with instructions. The code looks reasonably structured and commented.

I did not install and run as I do not have a Matlab license right now.

Evidence of a cascading positive tipping point towards electric vehicles

Referee Report

The manuscript centers on the determination of a “tipping point” for the adoption of EVs in specific lead country markets and globally. To test the relevant hypotheses, the authors use a comprehensive dataset over the composition of car sales between 2016 and 2023, tracking 2,452 models in 33 countries. The authors find that several European markets and China have tipped to EVs, although this is not the case for the US. EV sales are rising exponentially while those of ICEVs are declining. The data and methods of the manuscript predict that the tipping point likely lies within the next few years in lead markets of the EU and China, and potentially the US; thereafter, the tipping point could spill out into peripheral vehicle markets across the rest of the world.

As a measure of the loss of resilience of the incumbent technology, the authors find that the variance and lag-1 autocorrelation of fluctuations in ICEV sales increased beforehand, consistent with the approach to a tipping point. For the purpose of identifying the conditions that can accelerate the convergence to tipping points, the paper uses simulations of technology evolution to identify timescales for cost-parity and policy frameworks that could accelerate the transition to largely eliminate ICEVs before 2050.

The paper is well motivated and generally clear. Two revisions would strengthen the presentation and research project.

The description of the method is unnecessarily disparate and less integrated than possible. In either the Introduction of Content and Theory section, the authors should organize a single paragraph describing the sequential steps used to identify a “tipping point.” Seemingly, each step corresponds to a section/figure in the Appendix that can be referenced. To the extent that each step/stage functions sequentially or in parallel interactively, it may also help to create a diagram that exhibits the transition from policy to tipping point to EV dominance.

The second point is more fundamental; it concerns the basic driver of the transition. Succinctly put, arguably the convergence of market phenomena – the experience curve, cost-cutting technological innovations, networking scale economies, etc. – have all been the causal consequence of a single regulation, i.e., the phase out of ICEVs by 2035.

L 33,34: Plug-in hybrid electric vehicles (PHEVs) overcome issues of range anxiety perceived with EVs, but they are now to be phased out along with ICEVs in the EU and the UK by 2035,...

The zero quota restriction has functioned as an exogenous shock, requiring the conditions for the endogenous evolution of the “tipping point.” The zero quota alone stands as the sufficient condition for the market conditions that have ensued. The subsidies make the time-limited transition more feasible and politically palatable. The concern is that the presentation gives too much weight to the independent force of the markets, while in fact, the behavior of the markets

has been shaped by a drop-dead deadline that has made a “tipping point” and the market adjustments inevitable. The story is about how markets adapt to a zero quota over a fixed time horizon, a process that involves a “tipping point.” But this then raises the issue of the precise meaning of “tipping point” and the 7 references to a “self-propelling” transition.

L. 39,40: Hence it is important and timely to examine whether the observed diffusion of EVs could become self-propelling and tip towards a state of dominance.

The meaning of and conditions for the transition to be “self-propelling” need clarification. Does the term mean that the 2035 zero quota and subsidies could be eliminated, still resulting in the convergence to 100% EVs, or does it mean that given the existing body of government policy and subsidies, the adjustment process requires no further government initiative, i.e., the remainder of the transition can be achieved through the fixed set of public-policy conditions prices and incentives. Status quo vs. full market transition with neutral government policy is a critical distinction that is made ambiguous by the term “self-propelling”.

It appears that the authors have the data and methods to determine or offer informed speculation regarding the two alternative tipping-point and self-promotion scenarios. Even if the scenario is that of status quo policy-driven transition in the EU and China, the spillover effects to the rest of the global market (now including the US likely w/o such direct incentives) is a critical feature of the research.